# Influence of Plasma-Activated Water on Physical and Physical–Chemical Soil Properties

**Jana Šimečková [1,\*], František Krčma [2,\*] , Daniel Klofáč [1], Lukáš Dostál [3] and Zdenka Kozáková [2,\*]**

[1] Faculty of AgriScience, Department of Agrochemistry, Soil Science, Microbiology and Plant Nutrition, Mendel University in Brno, Zemědělská 1, 61300 Brno, Czech Republic; xklofac@node.mendelu.cz

[2] Faculty of Chemistry, Institute of Physical and Applied Chemistry, Brno University of Technology, Purkyňova 118, 61200 Brno, Czech Republic

[3] Faculty of Electrical Engineering and Communication, Brno University of Technology, Technická 10, 61600 Brno, Czech Republic; dostall@vutbr.cz

\* Correspondence: jana.simeckova.uapmv@mendelu.cz (J.Š.); krcma@fch.vut.cz (F.K.); kozakova@fch.vut.cz (Z.K.)

**Abstract:** Recently, the bactericidal and fungicidal effects of plasma-activated water (PAW) have been confirmed for its application in agriculture. Although the PAW application is beneficial in plant growth, no information is available about processes induced by PAW in soil. This paper gives the first experimental results about PAW's influence on selected physical and physical–chemical properties of soil. PAW was prepared using the dielectric barrier discharge (DBD) operating in the multistreamer mode at a frequency of 11 kHz. The total energy consumption was 60 J/ml. The obtained results show minimal changes in the natural water evaporation from the soil exposed to PAW, slower tap water absorption if a higher amount of PAW (16 doses per 10 ml to 90 g of the soil) is applied, as well as water retention in the soil of over 30%. The soil pH remains in the neutral range of values even at the highest applied PAW amount of 1.7 weight of soil, which represents the best conditions with respect to the plant growth. Thus, we can conclude that the PAW application, even at high amounts, has no negative influence on the physical and physical–chemical properties of soil and it can be safely applied in sustainable, environmentally friendly agriculture.

**Keywords:** plasma-activated water; dielectric barrier discharge; reactive oxygen species; reactive nitrogen species; soil pH; water retention; water absorption; plasma agriculture

---

## 1. Introduction

Recently, non-equilibrium low temperature plasma (also called "cold plasma") generated at the atmospheric pressure has started to be a hot topic because of its nearly ambient neutral gas temperature that allows the interaction of plasma with temperature-sensitive and soft materials like polymers or even living tissues. Plasma state is composed of charged particles (electrons, positively and negatively charged ions), excited atoms and molecules, radicals, metastables, vacuum ultraviolet (VUV) and ultraviolet (UV) photons [1,2]. Therefore, plasma forms a very reactive, but cold environment and thus opens a possibility for treatments of various types of substances, both solid and liquid. Special attention is given to plasma interactions in and with liquids [3,4]. Common applications of plasma in this field are focused mainly on waste water treatment [5–8], hydrocarbons re-forming [9], the production and treatment of nanoparticles [10–12], surface treatment of solid materials [13,14] and, recently, also the utilization of so called plasma-activated water (PAW) for bio applications in medicine [15–18] and agriculture [19].

It is well known that plasmas initiate gas phase chemical processes leading to numerous chemically active products, the properties and identity of which may be well controlled by varying plasma parameters [1,4,20]. When air plasma is created over a water column (or in the bulk water or in gaseous bubbles in the bulk water), some of the plasma species formed in the gaseous phase above the water column are transported through the plasma–liquid interface into the water column. Thus, this water becomes activated and is called plasma-treated water or plasma-activated water (PAW) [4,21]. PAW contains significant amounts of chemically active species produced in plasma and at the plasma–liquid interface, especially reactive oxygen (ROS) and reactive nitrogen (RNS) species, together known as RONS. Some of the most important species appearing in the bulk liquid of PAW that might be involved in triggering cell mechanisms are $\cdot OH$, $\cdot O$, $NO$, $NO_2$, $\cdot H$, $H_2O_2$, $O_3$, $NO_2^-$, $NO_3^-$, and $ONOOH$ [20]. Changes of chemical composition also lead to the acidification of PAW. The extent of acidification and concentration of RONS is dependent on a variety of parameters such as the plasma discharge type, discharge power density, formed plasma volume, used electrodes, activation time, initial water chemistry and water volume [4].

The plasma-generated RONS coupled with the pH of the PAW are responsible for unique PAW properties, especially higher oxidation potential, that are efficient in the destruction of organic pollutants, biological or chemical decontamination of air, water and soil as an alternative to classical pollution control techniques. High removal efficiency of a large variety of water contaminants (including phenolic compounds, organic dyes, pharmaceuticals, pesticides, etc.) can be achieved without requiring the addition of other chemicals, as in other advanced oxidation processes. The high oxidation potential of RONS mentioned above is also responsible for PAW's strong antibacterial and fungicidal properties [17,18]. Recently, the application of PAW in agriculture as an alternative to commercial fertilizers has become a focus of researchers' interest [19].

Nitrogen is a vital plant nutrient, which plays a major role in the development of dark green leaves, and promotes growth of roots, stems, and leaves of the crop, eventually leading to an increased yield [22]. Production of conventional nitrogen fertilizers requires a substantial supply of fossil fuels. A recent study estimates that almost 1.8% of the global fossil fuel consumption is used for ammonia production [23]. The use of fossil fuels to produce nitrogenous fertilizers is a significant contributor to global warming and climate change. At the farmland, the nitrogen fertilizer, which is typically available in a granular form, is applied (typically sprinkled) to the soil as a shot dose. Urea and ammonium nitrate are the most common forms of nitrogen fertilizers, which are converted by the bacteria in the soil into plant-consumable nitrates. It has been proven that almost 50% of the added nitrogen is converted into this plant-consumable nutrient [24]. The rest terminates in water and contributes to the surface and underground water contamination.

Based on all these consequences related to the conventional nitrogen fertilizers, it is necessary to find alternative sources of nitrogen applicable in plant nutrition. The use of PAW, which is rich in nitrogen content, seems to be a promising alternative to the conventional nitrogen fertilizers. Additionally, PAW can be also used for fertilization through the plant leaves. In such cases, PAW's antibacterial and fungicidal properties are beneficial, too. Further, the whole process of PAW production by the application of low temperature plasma on water is kept at high bio quality, i.e., without any hazardous chemicals added into the treated water and by using natural resources only.

During recent years, researchers have focused on PAW generation using different methods [25]. For example, a DBD plasma source was used for PAW generation from demineralized water, which was subsequently successfully applied in the seed germination and plant growth of tomato, pepper and radish seeds [26]. The DBD plasma system was also used for PAW generation using tap water and the positive effect on the root length of lentil seeds was observed, too [27]. Another research team has generated PAW from tap water using an atmospheric pressure plasma jet and observed four times higher root growth of lentil seeds in the first 15 days [28]. Plasma systems using a spark discharge and a transferred gliding arc discharge were also studied to produce PAW from tap water and apply it within growth studies on watermelon, radish, tomato, and pole bean plants [29].

Besides the PAW function as the source of nitrogen for enhanced plant growth, a lot of studies have been realized to see anti-microbial effects of PAW on different microorganisms. Researchers have shown that PAW can inactivate, kill or even destroy a wide range of microorganisms like bacteria, fungi, biofilms, viruses and spores [17,18,30]. The loss in membrane integrity is believed to be caused by the generation of ROS and the resulting lipid peroxidation, showing that the whole cell wall of the bacteria breaks down after treatment with PAW.

A synergetic effect of PAW as both environmentally friendly fertilizer and sterilization medium seems to be very promising for its application in agriculture. However, a general problem for its broad use in agriculture is completely missing information about its influence on soil and its properties. Although PAW is applied in the form of spray on leaves, some amount falls down and it is absorbed by the soil where it can induce positive or negative changes of its properties. The lack of this knowledge is the motivation for the presented study. One of the most important soil properties monitored by agronomists is water retention referring to the mechanisms and processes related to changes in soil water content versus its energy status. It comprises the amount of water held in soil and the potential energy with which the water is held [31]. Physical properties define movement of air and water/dissolved chemicals through soil, as well as conditions affecting germination, root growth, and erosion processes [32]. Physical–chemical soil properties combine both physical properties such as aggregation, and soil water holding capacity, and chemical properties of soil (e.g., pH) [33]. Physical–chemical properties of soil are usually considered as indicators of soil quality [34].

The paper brings the first results obtained by the application of PAW on selected soil samples. PAW was produced from distilled water using the DBD plasma system. Plasma diagnostics were carried out by optical emission spectroscopy in order to determine the active species in plasma. Treated water was analyzed from both physical (pH, conductivity) and chemical viewpoints (concentration of nitrogen oxides and hydrogen peroxide). The evaluated soil properties were water evaporation between individual PAW applications, repeated water absorption and water holding in the soil. From physical–chemical soil properties, the influence of PAW on the soil pH value (with respect to distilled water and potassium chloride [35]) was analyzed.

## 2. Materials and Methods

The PAW was prepared in a dielectric barrier discharge (DBD) plasma system schematically shown in Figure 1. The system consisted of a bottom part of a Pyrex glass Petri dish (90 mm in diameter, thickness of 1 mm, dielectric constant of 4.8) with a graphite outer electrode manufactured using graphite lacquer (Cramolin, Kissing, Germany) in the central position (50 mm in diameter). The other electrode was made of an alumina ceramic plate $102 \times 102 \times 0.7$ mm$^3$ (dielectric constant of 9.8) with a PVD manufactured upper silver electrode of the same dimensions as the graphite electrode. This electrode was put just on the Pyrex glass Petri dish top. Both electrodes were connected to the power supply (Lifetech) operating at a frequency of 11 kHz. The power supply was constructed symmetrically, so no electrode was grounded. The sinusoidal peak to peak voltage was 16 kV. A quantity of 75 mL of distilled water was treated 8 times for 15 s. This water volume ensured a gaseous gap of 3.2 mm between the distilled water surface and the upper alumina electrode. Between treatments, the system was opened to ventilate the exhaust gas mixture and introduce the fresh ambient air. Total energy supplied from the electrical network was ($36 \pm 2$) W. The PAW amount needed for each application was prepared immediately before its use. Because of the limited volume of the plasma rector, the prepared PAW was homogenized by mixing of all doses together to decrease the PAW quality dispersion. Distilled water was kept in the same room as the soil samples, so it had the same temperature of ($21 \pm 1$) °C during the whole experiment. The PAW temperature at the moment of application was about 2 °C higher than the laboratory temperature.

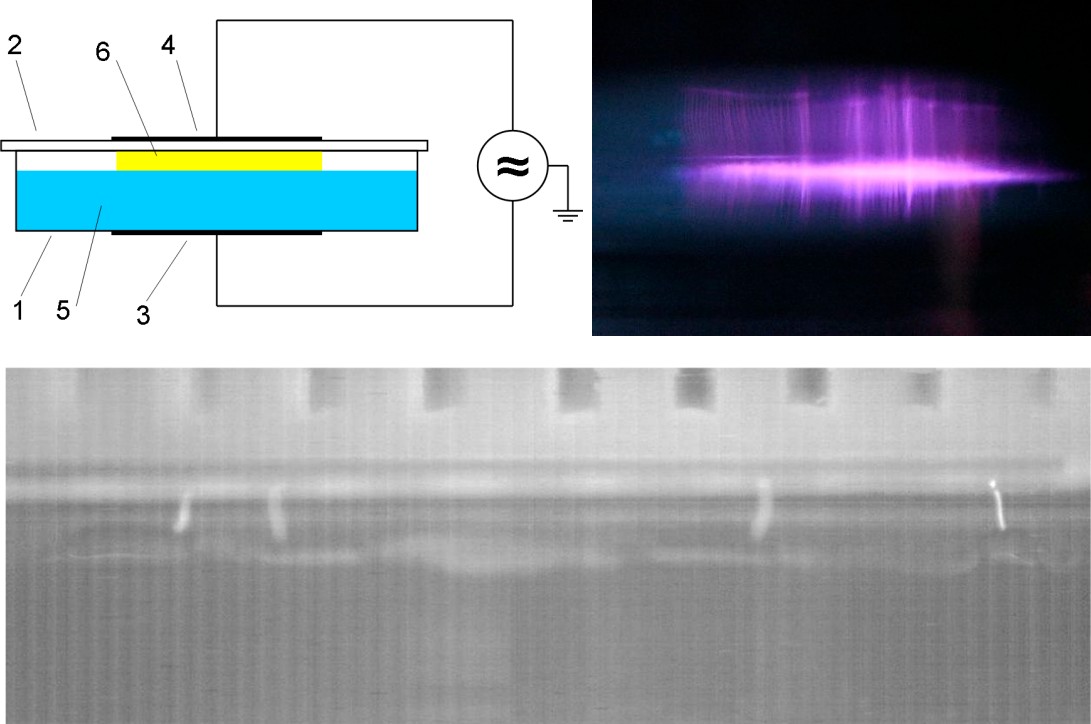

**Figure 1.** Upper left: Scheme of the PAW production experimental setup. 1—Pyrex glass Petri dish; 2—alumina plate; 3—graphite lacquered electrode; 4—silver electrode; 5—treated water; 6—plasma zone. Upper right: Photo of plasma during the discharge operation in the reactor. Bottom: Discharge channels visualized by the fast camera (movie, see Supplementary Materials. Acquisition rate was 1000 frames per second, projection rate was 2 frames per second).

The plasma itself was characterized by optical emission spectroscopy using the Jobin Yvon TRIAX550 (currently Horiba, Longlumeau, France) spectrometer with the $LN_2$ cooled CCD detector. Light emitted by the discharge was collected by the lens with the focal length of 150 mm to the multimode optical fiber. The 1 cm width quartz window was installed as a part of the Pyrex glass Petri dish in order to observe radiation below 300 nm. The 1200 gr/mm grid was used for the current experiment; grating with density of 3600 gr/mm was used for OH and NO measurements. Overview spectra were recorded in time averaged mode; selected peaks were also measured in time evolution, both with integration time of 30 s. The fast camera Photron FASTCAM SA-X2 (Photron, San Diego, CA, USA) was used for the discharge visualization. The time resolution of 1 ms was used in the current study.

The PAW physical–chemical properties were characterized by pH and conductivity measurement using the WTW InoLab (WTW, Prague, Czech Republic) and GRYF 107 L (GRYF HB, Havlíčkův Brod, Czech Republic) apparatus, respectively. Concentration of hydrogen peroxide was determined calorimetrically using the titanium reagent [36]. Concentrations of nitrites and nitrates were measured using the commercial $NO_x$ test kits (TetraTest $NO_2{}^-$ and TetraTest $NO_3{}^-$) that are based on the Griess reaction [37].

The soil samples were prepared using homogenized soil (the soil type fluvisol [38], the soil texture sandy loam, localization 48°59′10′′ N, 16°37′43′′ E) that was sieved using a 2 mm sieve and stabilized at ambient laboratory conditions (temperature of (21 ± 1) °C and relative humidity of (20 ± 3)%) for 1 week. The Kopecky´s rolls (inner diameter of 52 mm, height of 47 mm; for additional information see the Supplementary file) covered at the bottom by a stainless steel textile (mesh of 0.415 mm, wires 0.22 mm) were fed by 90 g of soil and compressed homogeneously by 500 kPa for 60 s using

the RYA-RAN Injection Moulding Press for Test Samples (Tyco Electronics AMP, Barcelona, Spain). The final soil density before the experiment beginning was 1.27 g/cm$^3$.

Samples were divided into two groups. The distilled water (DW) was applied in the control group; PAW was used in the other. Three identical samples were used at each set of conditions to obtain the relevant data. Nine series of samples were evaluated in the current study. One series served as the reference; the others differed in a number of PAW applications to determine the influence of a long term PAW application on the soil properties. Every series was carried out in two conditions.

Each sample (each Kopecky´s roll) was always weighed before (to observe evaporation) and after (to control the applied dose) application of water. Water was applied on the soil surface in the amount of (9.87 ± 0.17) g (exactly measured by the change of the sample weight) in the 48-h interval using a pipette with a modified output to avoid formation of craters on the soil surface. After the last watering of both series (1, 2, 3, 4, 6, 9, 12 and 16 doses of distilled water and PAW, the 48-h interval action of applied water was maintained), the first group of samples (containing 3 Kopecky's rolls) was put in the vacuum dryer (Binder VD53, BINDER GmbH, Tutlingen, Germany) at laboratory temperature; the second group was dried by laboratory air.

Repeated water absorption of samples (the first group of three Kopecky's rolls) was tested by saturation on the glass covered by a filter paper with continuous tap water delivery (Figure 2 left; for tap water composition, see Table 1). The absorption was measured at selected times by sample weighing. In next step, the fully saturated soil samples were put on the dry filter paper (4 layers) to monitor water holding capability (Figure 2 right). The water holding capability was measured at selected times by sample weighing; each sample was put on a new pack of dry filter paper sheets after the weighing.

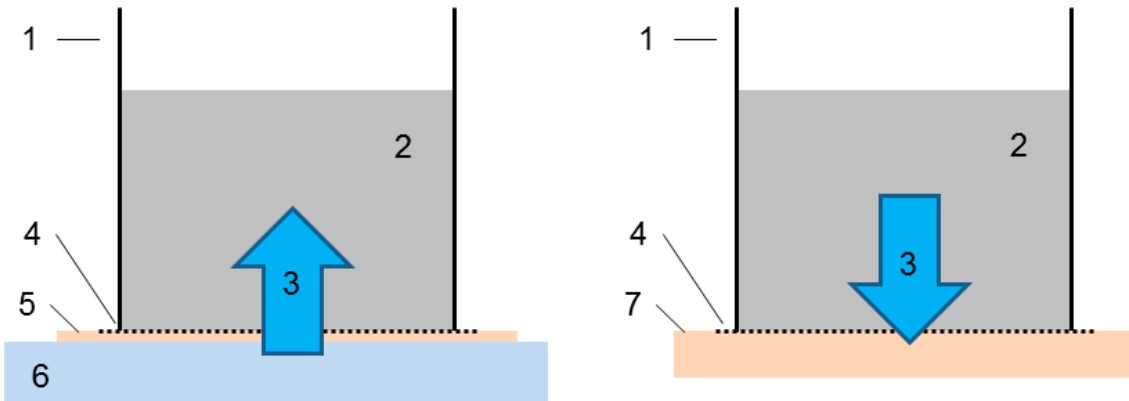

**Figure 2.** Scheme of water absorption (left) and water holding (right) experiment. 1—Kopecky's roll; 2—soil; 3—direction of tap water flow; 4—stainless steel mesh; 5—filter paper; 6—tap water reservoir; 7—4 layers of dry filter paper.

**Table 1.** Tap water composition [39].

| Quantity | Value |
|---|---|
| Conductivity | 450 μS/cm |
| pH | 7.49 |
| Total hardness | 1.96 mmol/L |
| Iron | <0.03 mg/L |
| Ammonium ions | <0.03 mg/L |
| Nitrates | 26.50 mg/L |
| Nitrites | <0.00 mg/L |
| Chlorides | 17.20 mg/L |
| Total oxygen content | 1.51 mg/L |
| Free chlorine | 0.12 mg/L |

The Kopecky´s rolls with soil for the pH value measurements (the second group of samples, consisting again of three Kopecky´s rolls) were stored at ambient laboratory conditions up to their processing. The soil was removed from the roll individually and finally dried at laboratory conditions for 24 h on the filter paper. An amount of 10 g of the soil was shaken with 25 mL of boiled distilled water or with 25 mL of 1 M solution of potassium chloride for 1 min. The colloid solution was kept in a closed vessel for 24 h at laboratory temperature. After that, the solution was shaken again for 1 min and solution pH was measured using a flat membrane electrode Metrohm (Metrohm Česká republika s.r.o., Prague, Czech Republic). The pH values measured using distilled water reflect free proton concentrations in the soil, while pH values measured using the KCl solution also reflect bounded protons in the sorption system of the soil [35].

## 3. Results and Discussion

The DBD is operating in the filamentary mode with the simultaneous appearance of more streamers that are unstable in space and time, as shown in Figure 1 and in the movie presented in the Supplementary Materials. The discharge emission spectrum was collected to obtain information about active species that can pass through the plasma–liquid interface into the water. An example of the measured data is shown in Figure 3. The band of the nitrogen second positive system ($N_2$ $C^3\Pi_u \rightarrow B^3\Pi_g$) is dominant in the UV-blue region of the spectrum, the nitrogen first positive system ($N_2$ $B^3\Pi_g \rightarrow A^3\Sigma_u^+$) is the main emission in the red-NIR region. Besides them, emission of the nitrogen first negative bands ($N_2^+$ $B^2\Sigma \rightarrow X^2\Sigma$) originating at the ground vibrational level and OH (A $A^2\Sigma \rightarrow X^2\Sigma$) 0-0 band were measured in the UV part of the spectrum. Hydrogen (656 nm) and oxygen (777 nm) atomic lines were identified, too. The NO radical spectrum was recorded separately in the UV region.

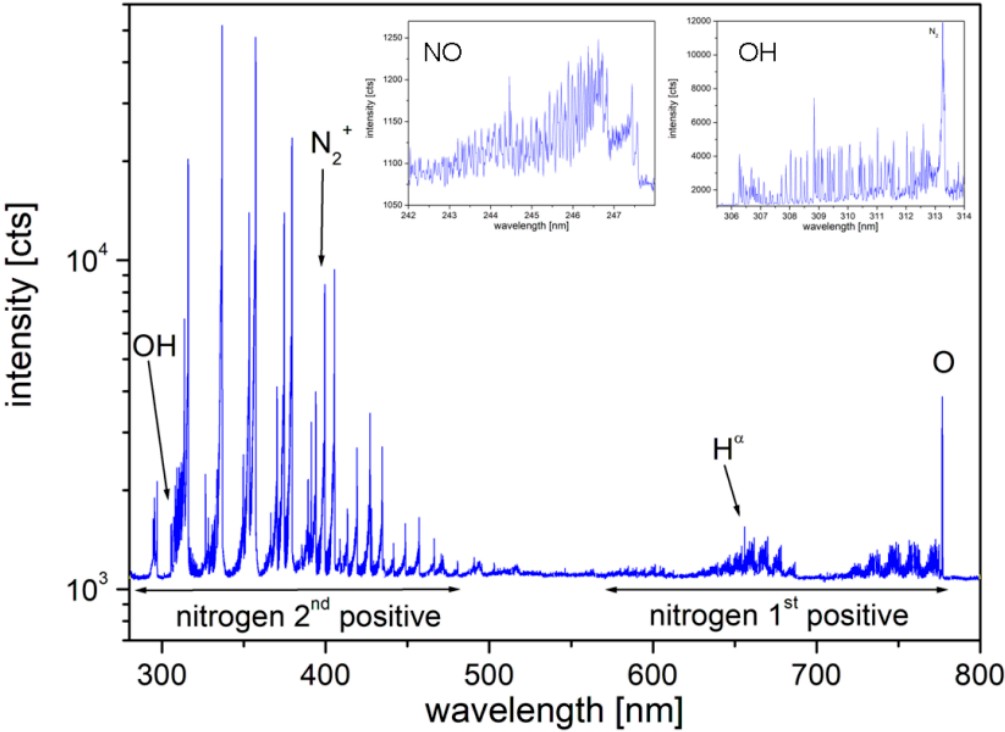

**Figure 3.** Example of the discharge emission spectrum.

Based on the recorded spectra, the rotational temperature of (810 ± 50) K was calculated from the intensities of the lowest five OH (A-X) 0-0 band transitions by the Boltzmann plot method (for details, see [40,41]). Vibrational temperature of (1670 ± 100) K was calculated from the −2nd sequence of the

nitrogen second positive system using the same method (for details, see [36,37]). These results confirm that the discharge forms non-equilibrium plasma.

The physical–chemical properties of PAW that we observed within this study were conductivity and pH. The conductivity of PAW prepared by the DBD from distilled water was enhanced to 34 µS and the pH value dropped to 6.7 from the value of 7.2 measured for DW. The chemical activity of PAW was evaluated by the determination of the produced amount of hydrogen peroxide, nitrites and nitrates. The average concentration of hydrogen peroxide in PAW was $(1.4 \pm 0.4)$ mg/L $((0.040 \pm 0.012)$ mmol/L), the concentration of nitrites was $(0.753 \pm 0.009)$ mg/L, and nitrates $(20.4 \pm 1.8)$ mg/L.

Both PAW and pure distilled water were applied in several doses = number of applications (up to 16 doses, each dose was $(9.87 \pm 0.17)$ g) on the soil samples in the 48-h interval. Subsequently, the total amount of evaporated water was determined by the sample weighing 48 h after the water application. Results are compared in Figure 4. At the lower number of applications (up to 6), the evaporated amount of water increased slowly and more or less independently on the water type. Before the water application, the soil humidity was in balance with the surrounding air humidity, which was 20%. Therefore, the soil samples absorb water intensively and prevent water evaporation until the soil is saturated. At higher numbers of application, estimated amounts of evaporated water increased more or less linearly with further water doses and a remarkable difference of 4% between PAW and DW was achieved. The experimental error of points in Figure 4 is less than 0.5%.

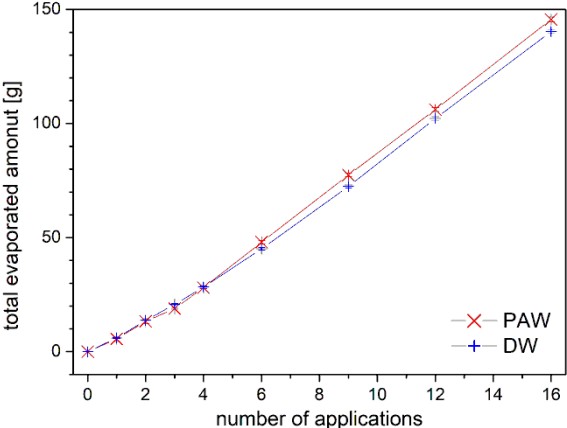

**Figure 4.** Evaporation of applied water during its applications (PAW—plasma-activated water; DW—distilled water).

The tap water absorption was measured using Kopecky´s rolls dried in the vacuum dryer (the first group of samples). Results obtained for all numbers of applications are shown in Figure 5 for DW and PAW separately. Figure 6 shows tap water absorption amounts in time dependence in the case of selected numbers of DW/PAW applications (3, 6, and 16, respectively). Results show that the amount of absorbed tap water was higher than in the case of the reference for the low number of DW/PAW applications. The first two applications of 10 ml were not sufficient to fill the whole roll volume because the experiment had started with a dry soil. Due to this fact, not all of the roll volume was modified by the DW/PAW application and its absorption was increasing with the number of applications. On the other hand, a higher number of applications led to soil water saturation just after the application and potential migration of soil microparticles that could modify soil porosity and a subsequent water flow through the soil. We suppose that soil has a slightly different wettability by DW and PAW and thus maximal absorption is at a higher number of PAW applications. The higher number of applications leads to a decrease in the tap water absorption capability and this decrease is similar for both DW/PAW, except the highest used number of applications, where PAW shows a significantly worse capability. There is no significant difference in the sorption capability in time during the first few hours if the lower number of DW/PAW applications is used (see Figure 6). However, the absorption is significantly

slower at the beginning if the high number of DW/PAW doses is applied. Other trends are the same as it was described above.

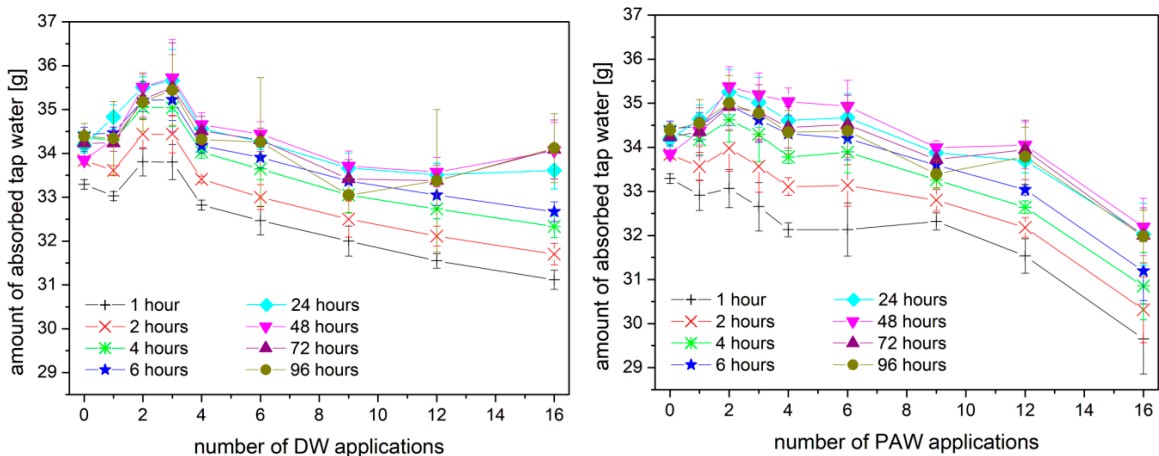

**Figure 5.** Absorption of tap water in dependence on the number of DW/PAW applications.

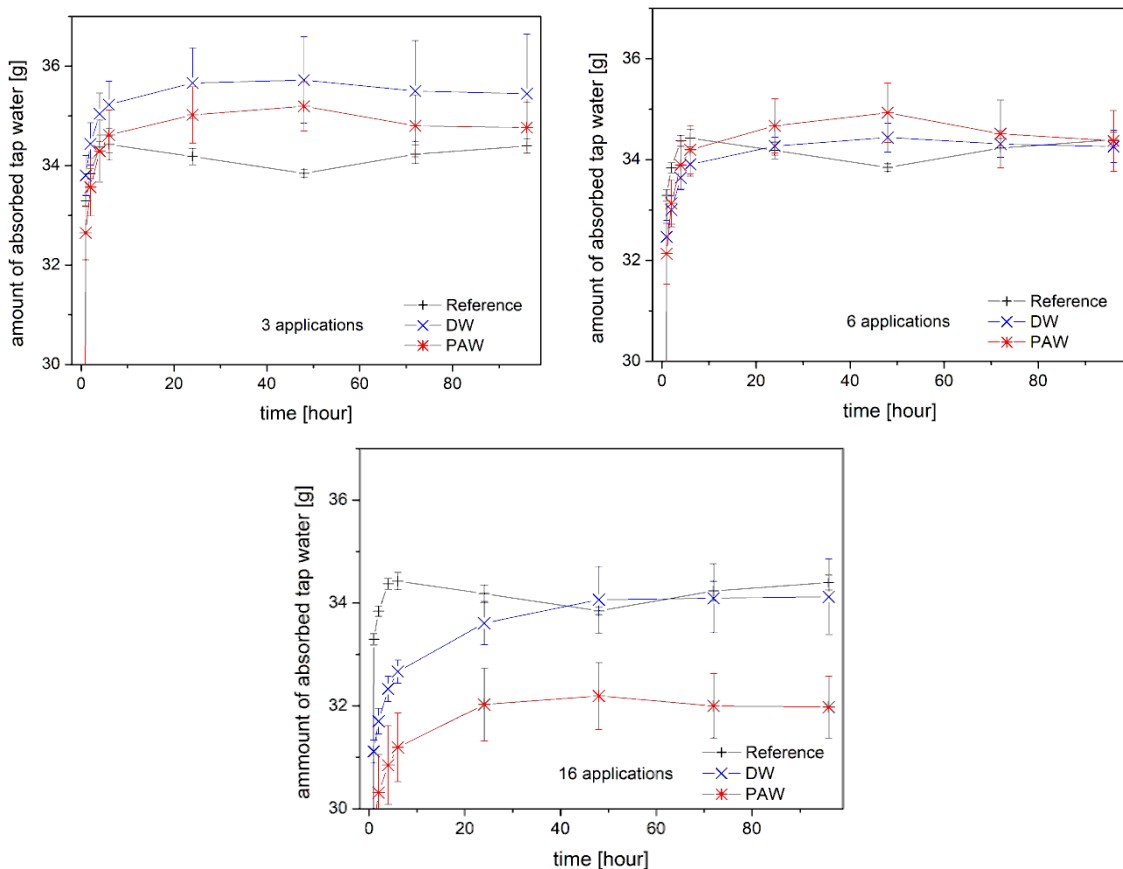

**Figure 6.** Absorption of tap water at the selected number of DW and PAW applications.

Tap water removal in dependence on the number of DW/PAW doses is shown in Figure 7 for the selected times. Results show that the increasing DW applications leads to the increase in removed water, so the water retention is decreasing. On the other hand, PAW doses had no effect on the soil retention. The higher tap water removal after two DW/PAW applications was probably again connected to the soil microparticles migration into soil capillaries and thus to the modification of the soil porosity. This difference is more visible in the time dependences shown in Figure 8. It is interesting that the

high number of PAW doses leads to the small retention improvement that is visible also at the lower number of doses. This effect might be related to the surface tension of the soil particles that might be influenced by the PAW application. It seems that this application preserves the porous structure of the soil while the distilled water application attracts ions and blocks the pores in the soil, which leads to its lower penetration ability. This can be connected to the potential changes in the soil microbiology that was not a subject of the presented study.

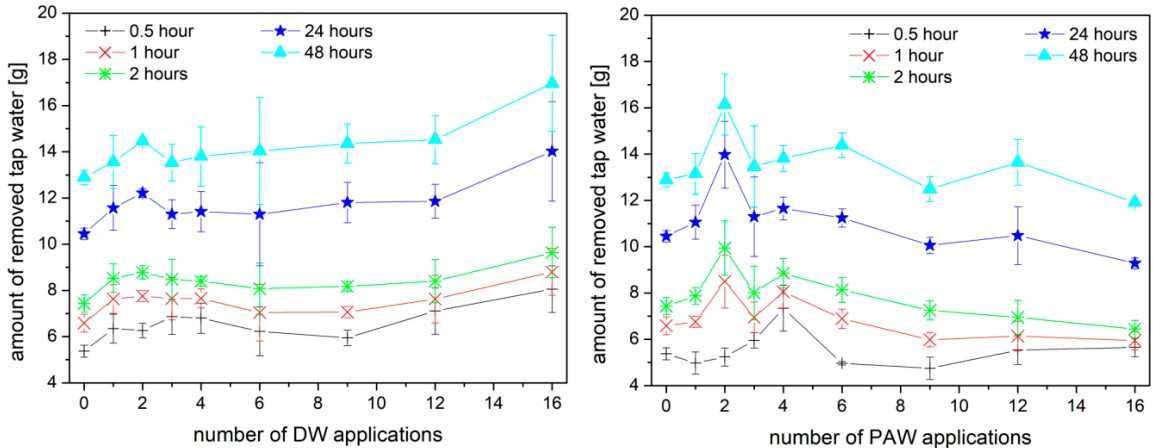

**Figure 7.** Removal of tap water in dependence on the number of DW/PAW applications.

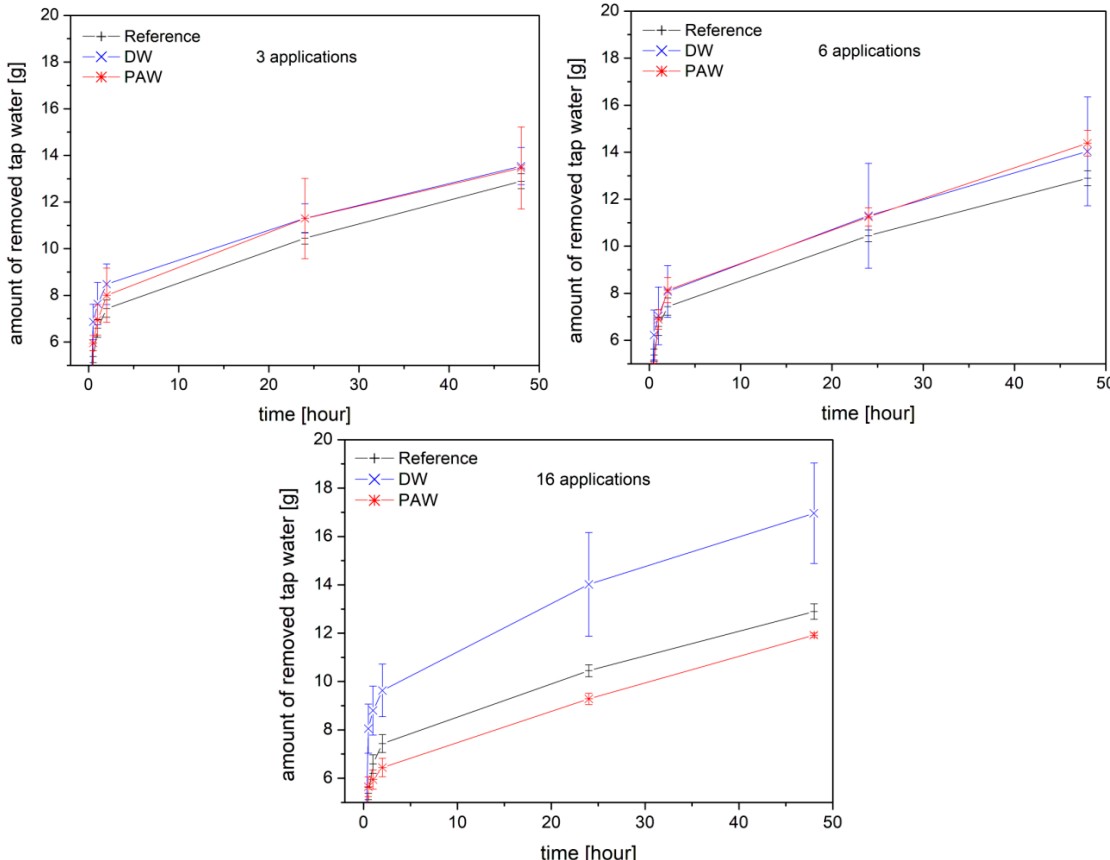

**Figure 8.** Removal of tap water at the selected number of DW and PAW applications.

The soil pH values were evaluated by two methods: the free proton concentration in the soil was measured in the distilled water (pH/$H_2O$) while the total amount of protons bounded also in the sorption system of soil was measured in the 1M KCl solution (pH/KCl). Obtained results from both methods are presented in Figure 9. The pH value determined for the reference soil sample was $7.31 \pm 0.03$ for the pH/$H_2O$ measurement and $6.44 \pm 0.02$ for the pH/KCl measurement, respectively. The lower pH value of the pH/KCl measurement indicates higher amount of hydrogen cations bounded in the soil sorption system.

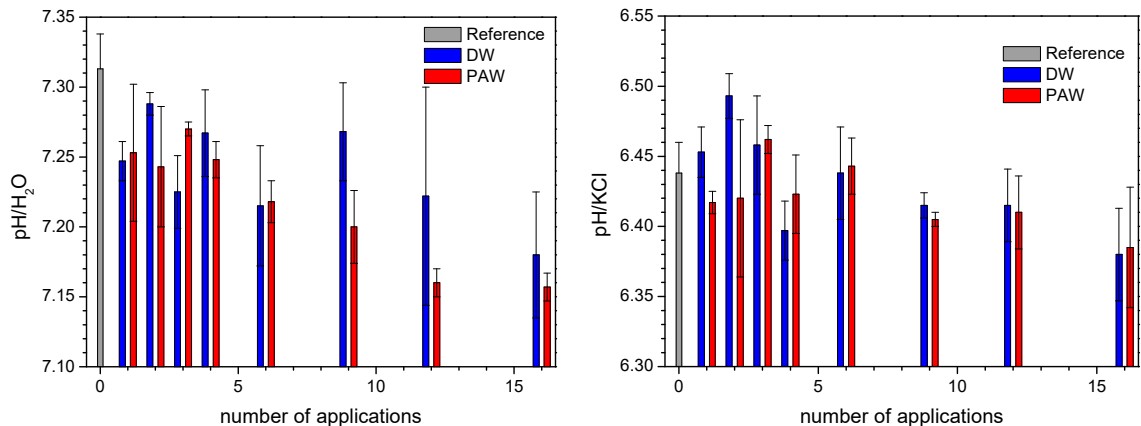

**Figure 9.** Soil pH values at the selected number of DW and PAW applications.

By the application of PAW that has a slightly acidic pH value (6.70), both pH values (representing free as well as bounded protons) were decreased by the increasing number of PAW applications. Although the pH decrease by the PAW application was evident (from $7.31 \pm 0.03$ to $7.16 \pm 0.01$ after 16 PAW applications), it did not reach acidic values that might change the physical–chemical balances in the soil. The application of pure distilled water led to only slightly decreased pH values.

An interesting phenomenon was observed for the pH/KCl values at the first two applications of distilled water because the pH values were enhanced. This effect might be explained by the low number of distilled water applications that prevent the release of hydrogen cation from the sorption system.

An important fact of the obtained results is that both pH measurements indicate a neutral soil reaction even at higher PAW applications [42]. Therefore, such conditions are still good for the plant growth.

## 4. Conclusions

The presented paper gives the first results about PAW application on the selected physical and physical–chemical properties of the soil. The PAW was prepared in the DBD system operating at 11 kHz with energy consumption from the electricity network of 60 J/ml. The discharge itself operated in the multistreamer mode with steamers unstable in space and time. The presence of reactive nitrogen and reactive oxygen species was confirmed by the optical emission spectroscopy of the discharge and by colorimetric measurements using selective reagents in the liquid phase.

PAW and distilled water as the reference were applied on the soil samples in multiple doses of about 10 ml per 90 g of soil in the 48-h interval. The natural evaporation during the PAW/DW applications was measured in the 48-h interval and it was observed that evaporation is about 4% higher in the case of PAW. The tap water sorption was measured during 96 h in dependence on the number of PAW/DW applications. A small number of applications led to better tap water sorption in both cases, but the very high number of applications led to lower sorption ability of the soil exposure to PAW; the difference with respect to DW was about 6%. The tap water retention was measured in the 48-h experiment. No significant changes were observed at the lower number of PAW/DW applications but

the high number of applications led to over 30% better retention in the case of the soil samples exposed to PAW with respect to the soil samples exposed by DW.

Finally, the soil pH changes were determined with respect to water and 1 M KCl solution. The first one determines free hydrogen cations, the second one also provides information about cations built up in the soil sorption system. The slight acidification increasing with the number of PAW/DW applications was observed in both cases with respect to water; no significant changes were observed with respect to the KCl solution. This means that both water applications increased the concentration of the free hydrogen cations but there was no remarkable change of their presence in the soil sorption system. The soil pH remained in the neutral range of values even after the highest used number of PAW/DW applications, so the soil was still in the best conditions for plant growing.

The obtained results showed that even very high PAW exposure had no significantly negative influence on the physical and physical–chemical soil properties. Thus, PAW can be a good candidate for application in sustainable, environmentally friendly agriculture. The influence on the soil micro and macro-organisms will be the subject of the next series of experiments.

**Supplementary Materials:** The following are available online at http://www.mdpi.com/2073-4441/12/9/2357/s1, Video S1: DBD operation.

**Author Contributions:** Conceptualization J.Š. and F.K.; methodology J.Š., F.K. and Z.K.; investigation J.Š., F.K., D.K., L.D. and Z.K.; writing—original draft preparation, and review and editing J.Š., F.K. and Z.K.; visualization L.D. All authors have read and agreed to the published version of the manuscript.

**Funding:** This research received no external funding.

**Acknowledgments:** The research was financially supported by the Internal Grant Agency of Mendel University grant No. AF-IGA2020-IP085.

**Conflicts of Interest:** The authors declare no conflict of interest.

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
