# Peer review of "Influence of Plasma-Activated Water on Physical and Physical–Chemical Soil Properties"

_water, doi:10.3390/w12092357_

Round 1

Reviewer 1 Report

Please find an attached reviewer's report. 

Author Response

Dear referee,

Many thanks for your really valuable comments. We tried to improve the manuscript according to all your recommendations. We hope that this revised version will be much better comprehensive for readers. The detailed response is done below point to point.

  • Title, Is a word ‘selected’ necessary?

A: Yes, this word is important because we did not study effects on soil aggregates formation that is important for plants growing.

In Abstract, tone down is necessary, ‘no information’, ‘the first’ and ‘no negative influence’ are not suitable for the paper.

‘Although the PAW application is beneficial in a plant growth, no information is available about processes induced by PAW in soil. This paper gives the first experimental results about PAW influence on selected physical and physical-chemical properties of soil.’  

Thus we can conclude that the PAW application even at high amount has no negative influence on the physical and physical-chemical properties of soil and it can be safety applied in the sustainable environmentally friendly agriculture.

A: The first two were removed; the last one was kept because it is important. We know that PAW has more effects on soil and not all of them can be positive (like potentially for the soil bacteria, that will be subject of incoming study).

  • Definition for couple of words is necessary for readers in different research field, for examples ‘water retention’ in agriculture, ‘physical soil property’, and ‘physical-chemical soil property’

A: Definitions were added into the Introduction with related references:

“One of the most important soil properties monitored by agronomists is water retention referring the mechanisms and processes related to changes in soil water content versus its energy status. It comprises the amount of water held in soil and the potential energy with which the water is held [31]. Physical properties define movement of air and water/dissolved chemicals through soil, as well as conditions affecting germination, root growth, and erosion processes [32]. Physical-chemical soil properties combine both physical properties such as aggregation, and soil water holding capacity, and chemical properties of soil (e.g. pH) [33]. Physical-chemical properties of the soil are usually considered as indicators of the soil quality [34].”

  • Keep abbreviations PAW for plasma activated water and DBD for dielectric barrier discharge through the manuscript when they used from the second time.

A: Both abbreviations were used in whole text.

  • Wrong word ‘spectrometry’ through the manuscript, optical emission spectroscopy is correct.

A: Corrected.

  • In section of Materials and Methods,

Thickness for the both Pyrex glass Petri dish and alumina ceramic plate is necessary because dielectric property is important information to understand DBD system. 

Give more details of applied voltage, waveform and amplitude.  

A: Information about thicknesses and dielectric constants of alumina plate as well as Pyrex glass Petri dish were added.

Information about voltage form and amplitude was added, too.

  • Petri dish afterward the use of Pyrex glass Petri dish, it should be the dish or use full name. Readers could be confused polystyrene Petri dish.

A: The full term was used at all appearances in the text to avoid potential incorrect meaning.

  • Line 145, is ‘without time resolution’ means ‘with time-averaged mode’?

A: Corrected.

  • Line 147-148, ‘The sampling frequency of 1000 frames per second’ This is simply ‘time resolution is 1 ms.’

A: Corrected.

  • Line 156, check the location. Probably missing E?

A: You’re absolutely right, E was missing. Corrected.

  • Line 158, Kopecky’s roll is the most important to investigate soil property, can authors provide photographs for supporting information?

A: The supporting information is added as a supplementary material to present this clearly for people out of the soil community.

  • Line 164, ‘the second one’ should be the other’?

A: Corrected.

  • Line 200, ‘penetrate’ should be ‘dissolved’?

A: Reformulated – …can pass through the plasma-liquid interface…

  • Lines 209-213, Rotational temperature is considered the gas temperature because of the rapid rotational relaxation through inelastic collisions between molecules and atoms. The rotational temperature of 810 K (~537 degree C) was estimated. Author can give more information of temperature of PAW and DW as well. If possible, it seems that the measurement of soil temperature after watering with PAW and DW is valuable?

A: PAW was prepared in batches of 75 ml, for each application at least two batches were prepared. We measured PAW temperature of the batch just after generation. It was 4 degrees higher than before the plasma treatment and it was further cooled before its application. So it was not more than 2 degrees hotter than the soil. Distilled water was kept in the same room as all experiments were done, so it has temperature of about 22 degrees during the whole experiment. Thus the soil temperature was not changed by the application of DW or PAW within the experimental error. Information about temperature was added into the text.  

  • Line 216, confirm the conductivity to be 34 S and give pH value of DW, before the exposure of plasma.

A: This was incorrect copying; symbol micro was transferred to some other. It was corrected and pH value of distilled water was added.

  • Lines 218-219, Authors measured H2O2 (14 mg/L), NO2- (0.75 mg/L) and NO3- (20 mg/L). These molecules are stable; thus, they can be composed with chemicals. If do the same experiment with chemicals, authors will have the same results?

A: We have not realised such experiment, yet, but we do not suppose the same effect by the addition of pure H2O2 or NOx compounds. Commercial hydrogen peroxide contains stabilizers preventing spontaneous decomposition that might influence subsequent effect. Concerning addition of NOx salts, it might introduce further elements in a form of ions that could substantially change ionic strength of water. Moreover, PAW contains also peroxynitrite and other less stable species with lifetime of minutes that should be taken into account.

  • Lines 232-242, Why the tap water absorption is only different at 16 application? What kind of reasons are considerable for this result?

A: The trend of water absorption is also different at lower number of applications, but with only a negligible effect which is recognizable just after high amount of applications. Therefore, it has a cumulative tendency.

Results are discussed in the text before Fig. 7.

  • Lines 265-271, Why pH values are different even same sample but different pH measurement method? Which method is right to measure for this case?

A: There are two methods for the pH measurement: the first one (pH/H2O) gives information about free hydrogen in the solution while the second one (pH/KCl) represents also hydrogen that is bounded in the soil sorption system (that is why this value is lower than the first one). Both values are important because only after its comparison we can assume on changes in the soil sorption system.

  • Lines 297-298, … evaporation is about 4% higher in case of PAW. Is it meaning? It seems in error range.

A: Each experiment was carried out three times independently and the difference between these measurements was negligible (error bars in the graph are not depicted because they are below 0.5 %). Due to this fact, the 4% difference is significantly higher than the experimental uncertainty. Information about uncertainty was added into the text.

  • Line 303, 30% better retention, where from the value?

A: Information was added. It is with respect to the samples exposed to DW.

  • Line 310, visible, authors mean significant?

A: Yes, corrected to remarkable.

  • Line 312, best, authors mean suitable?

A: There are more levels of pH soil quality with respect to plants. The obtained data relates to the best pH value interval. Details see in reference [42]. 

Reviewer 2 Report

The paper deals with an important issue and the results would be welcome by the plasma agriculture community. The paper needs some clarifications before  publication.

The plasma system used makes possible the creation of active species in the treated liquid in reasonably low concentration. How the nitrite and nitrate concentrations reached compare with the nitrogen needed by the plants and provided by the fertilizers. Why authors have chosen this particular plasma system? For example the surface-wave microwave discharges can provide order of magnitude higher concentrations.

Figure 4 and 6, more critically 4, are very difficult to read. Please improve by changing and increasing the symbols.  

It is not clear what the authors mean by doses of liquids in the experiments. Does it mean that different amount of liquids are applied at once, so each dose is different, or it means the number of application at different time intervals of the same amounts.

It is also not clear related to Figure 4 -7, if the method is to apply different amount of PAW and DW to the soil to pre-treat and after that to apply tap water until saturation and that is presented on the figures. And why this protocol? Related to Figure 5, what is the effect that leads to lower absorption in the case of PAW?

Author Response

Dear referee,

Many thanks for your really valuable comments. We tried to improve the manuscript according to all your recommendations. We hope that this revised version will be much better comprehensive for readers. The detailed response is done below point to point.

  • The plasma system used makes possible the creation of active species in the treated liquid in reasonably low concentration. How the nitrite and nitrate concentrations reached compare with the nitrogen needed by the plants and provided by the fertilizers. Why authors have chosen this particular plasma system? For example the surface-wave microwave discharges can provide order of magnitude higher concentrations.

A: You’re right that there are many different plasma systems. Idea of this one is in its simplicity, scaleability and suitable price. This is the first study. We plan to compare different plasma-liquid systems (including continually operating) as well as different soils for future experiments under Plasma Agriculture COST Action.

  • Figure 4 and 6, more critically 4, are very difficult to read. Please improve by changing and increasing the symbols.

A: Symbols in Figures were changed according to recommendations.

  • It is not clear what the authors mean by doses of liquids in the experiments. Does it mean that different amount of liquids are applied at once, so each dose is different, or it means the number of application at different time intervals of the same amounts.

A: Doses = number of (water) applications means different number of the same water amount. We have tried to make it clearer in the text.

  • It is also not clear related to Figure 4 -7, if the method is to apply different amount of PAW and DW to the soil to pre-treat and after that to apply tap water until saturation and that is presented on the figures. And why this protocol? Related to Figure 5, what is the effect that leads to lower absorption in the case of PAW?

A: There are two physical parameters of soil observed in our work: 1) water absorption, and 2) water holding capacity. In both cases, soil was loaded by different doses of PAW or DW as a control. Then, all samples were dried in the vacuum dryer. Finally, the samples were tested for method 1) or 2) with tap water. Figure 2 was added to show the principle of both methods. The results are discussed in the text.

Round 2

Reviewer 1 Report

From reviewer's previous comment regarding Title, Is a word ‘selected’ necessary?

Here is authors’ reply, 'Yes, this word is important because we did not study effects on soil aggregates formation that is important for plants growing.'

Again, reader can recognize the soil aggregates formation was not obtained in this study with the word ‘selected’? Reviewer kindly remind ‘selected’ is generally not suitable to use in the title. 

Author Response

Dear referee

We are not fully sure for this correction but we made it. Soil is very complex mater and we can polemize what is or is not important. You are right that details can be found in the text. So we omitted the word "selected" from the paper title.

Many thanks again for you really valuable comments. We will consider them also in our further research.

Frantisek Krcma on behalf of all authors